# Enhancing sludge dewaterability in sequential bioleaching: Degradation of dissolved organic matter (DOM) by filamentous fungus *Mucor* sp. ZG-3 and the influence of energy source

Zhenyu Wang[1]*, Wen Feng[2], Shigang Tang[1], Jing Zhao[3], Guanyu Zheng[4], Lixiang Zhou[4]

1 College of Ecology, Lishui University, Lishui, Zhejiang, China, 2 Department of Soil Pollution Prevention and Control, Zhejiang Huanlong Environmental Protection Co., LTD, Hangzhou, Zhejiang, China, 3 College of Materials Science and Engineering, Henan Institute of Technology, Xinxiang, Henan, China, 4 College of Resources and Environmental Sciences, Nanjing Agricultural University, Nanjing, China

* 602681442@qq.com

**Data Availability Statement:** All relevant data are within the manuscript and its Supporting information files.

## Abstract

This study aimed to enhance sludge dewatering through sequential bioleaching, employing the filamentous fungus *Mucor* sp. ZG-3 and the iron-oxidizing bacterium *Acidithiobacillus ferrooxidans* LX5. The mechanism by which *Mucor* sp. ZG-3 alleviates sludge dissolved organic matter (DOM) inhibition of *A. ferrooxidans* LX5 was investigated, and the optimal addition of energy source for enhanced sludge dewaterability during sequential bioleaching was determined. Sludge dissolved organic carbon (DOC) decreased to 272 mg/L with a 65.2% reduction by *Mucor* sp. ZG-3 in 3 days, and the degraded fraction of sludge DOM was mainly low-molecular-weight DOM (L-DOM) which inhibited the oxidization of $Fe^{2+}$ by *A. ferrooxidans* LX5. By degrading significant inhibitory low-molecular-weight organic acids, *Mucor* sp. ZG-3 alleviated DOM inhibition of *A. ferrooxidans* LX5. In the sequential bioleaching process, the optimal concentration of $FeSO_4 \cdot 7H_2O$ for *A. ferrooxidans* LX5 was 4 g/L, resulting in the minimum specific resistance to filtration (SRF) of $2.60 \times 10^{11}$ m/kg, 40.0% lower than that in the conventional bioleaching process with 10 g/L energy source. Moreover, the sequential bioleaching process increased the sludge zeta potential (from -31.8 to -9.47 mV) and median particle size (d50) of the sludge particle (from 17.90 to 27.44 μm), contributing to enhanced sludge dewaterability. Inoculation of *Mucor* sp. ZG-3 during the bioleaching process reduced the demand for energy sources by *A. ferrooxidans* LX5 while improving sludge dewaterability performance.

## Introduction

Bioleaching emerges as a promising microbial approach for enhancing the dewatering process of sewage sludge, as evidenced by previous findings [1–8]. Within the realm of sludge bioleaching, the bio-oxidation of $Fe^{2+}$ by *Acidithiobacillus ferrooxidans* yields ferric flocculants capable of coagulating and flocculating sludge particles, thereby enhancing

**Funding:** This research was funded by Basic Public Welfare Research Program of Zhejiang Province (LGF21B070001), Lishui Public Welfare Technology Application Research Project (2021GYX13), and Science and Technology Innovation Project on Emission Peak and Carbon Neutrality of Jiangsu Province (BK20220040). The funders had no role in study design, data collection and analysis, decision to publish, or preparation of the manuscript.

**Competing interests:** The authors have declared that no competing interests exist.

dewaterability [9]. Concurrently, the hydrolysis of ferrous ions ($Fe^{3+}$) resulting from the bio-oxidation of ferric ions ($Fe^{2+}$) by *Acidithiobacillus ferrooxidans* releases $H^+$ ions, effectively reducing sludge pH. This acidification unequivocally promotes the solubilization of sludge-borne metals and contributes to the simultaneous improvement of sludge dewaterability [2, 3, 10–13].

The sequential bioleaching method involves the inoculation of the filamentous fungus *Mucor* sp. ZG-3 and *A. ferrooxidans* LX5 on day 0 and the end of day 1, respectively [14]. The performance of this method in sludge dewatering is superior to the conventional bioleaching process employing *A. ferrooxidans* alone, exhibiting a notable 25.9% reduction in specific resistance to filtration (SRF) [14]. SRF is widely used as a universal parameter for assessing sludge dewaterability, wherein a higher SRF value indicates inferior dewaterability [12, 15]. *Mucor* sp. ZG-3 enhances sludge dewaterability as well as degrades sludge dissolved organic matter (DOM), thereby diminishing sludge-related toxicity to *A. ferrooxidans* LX5 and enhancing the activity of *A. ferrooxidans* LX5 and sludge dewaterability [14].

The growth and activity of *A. ferrooxidans* are considerably impeded by DOM in sludge, primarily because of the inhibitory effects of sludge DOM on *Acidithiobacillus* species [16, 17]. However, filamentous fungi appear to exhibit suboptimal performance in degrading sludge DOM compared with other heterotrophic strains. For example, in a medium inoculated with *Brettanomyces* B65 and *A. thiooxidans* TS6, 83% of sludge DOM degradation can achieved after 36 h of incubation, whereas sludge DOM degradation reaches only 42.5%, yielding 390.85 mg dissolved organic carbon (DOC)/L after 3 days of fungal treatment [14, 18]. This concentration remains considerably higher than the inhibitory concentration of *A. ferrooxidans* LX5, as noted by Fang and Zhou [16], which stands at 150 mg DOC/L. Despite the persistence of elevated DOM concentrations, the inhibitory effect of sludge DOM on *A. ferrooxidans* LX5 is substantially alleviated [14]. Sludge DOM encompasses fractions of varying molecular weights (MWs), and different MW DOM fractions may exert differential effects on $Fe^{2+}$ oxidation by *A. ferrooxidans* LX5. The alleviation mechanism of sludge DOM inhibition on *Acidithiobacillus* species by filamentous fungi can be attributed to DOM degradation, primarily targeting inhibitory substances such as low-molecular-weight organic acids [14, 19]. However, conclusive evidence to corroborate this hypothesis is currently lacking. The impact of *Mucor* sp. ZG-3 on alleviating sludge DOM inhibition to *A. ferrooxidans* LX5 warrants further detailed investigation.

Studies and existing engineering practices have established $Fe^{2+}$ as a suitable and extensively utilized energy source for the growth of *Acidithiobacillus* species during the bioleaching process. Furthermore, some reports have examined the effect of $Fe^{2+}$ on sludge dewaterability in the context of conventional bioleaching process [1, 4, 9, 12, 20, 21]. For example, Wong et al. validated that a $Fe^{2+}$ concentration of 2 g/L optimally enhanced sludge dewaterability during conventional bioleaching with *A. ferrooxidans* LX5 alone, resulting in an 88% reduction in sludge SRF [12]. Given the superior efficacy of the bioleaching process in enhancing sludge dewatering performance, $Fe^{2+}$ supplementation for sequential bioleaching process must be optimized, with a view toward engineering applications and cost savings.

This study investigated the mechanism underlying the alleviation of sludge DOM inhibition to *A. ferrooxidans* LX5 by *Mucor* sp. ZG-3 and the optimal $Fe^{2+}$ supplementation for sequential bioleaching process. The primary objectives of this study were as follows: (1) to investigated the composition of the fraction of sludge DOM reduced by *Mucor* sp. ZG-3 and its toxity to *A. ferrooxidans* LX5 and (2) to evaluate the impact of $Fe^{2+}$ on sludge dewaterability and determine the optimal $Fe^{2+}$ supplementation during sequential sludge bioleaching process.

## Materials and methods

### Municipal sewage sludge sample

Municipal sewage sludge used in this study was obtained from the sludge-thickening pond of Lakou Wastewater Treatment Plant in Lishui City, Zhejiang Province, China. The plant treats $1.2\times10^5$ m$^3$/d of domestic sewage using an anaerobic-anoxic-oxic activated sludge process. The collected sample was transferred to the laboratory within one hour after sampling and then stored in a polypropylene container at 4°C for further use. Selected sludge properties, such as pH value (7.28), total solid content (3.62%), moisture content (96.38%), organic matter content (44.85%) and SRF ($8.71\times10^{12}$ m/kg), were immediately characterized after collection according to the respective standard method [22]. The moisture content was measured by oven-drying at 105°C, while the dried sludge sample was measured by oven-drying at 550°C for organic matter [22]. The sludge SRF was determined by using the Buchner funnel test [23].

### Preparation of biological inoculums

A filamentous fungus *Mucor* sp. ZG-3 (Genebank number KM668056) used in this study was isolated from municipal sewage sludge in our previous study [14, 24]. The optimal pH range for the growth of *Mucor* sp. ZG-3 was 4–8. *Mucor* sp. ZG-3 was incubated in 250 mL flasks containing 99 mL sterilized potato dextrose broth (PDB) with 1% (v/v) diluted spores suspension of *Mucor* sp. ZG-3 at a density of ~$1\times10^5$ spores/mL as the inoculum. The flasks were shaken at 120 rpm and 28°C for 48 h to prepare *Mucor* sp. ZG-3 inoculum [24]. Firstly, the whole *Mucor* sp. ZG-3 inoculum culture obtained above was filtered through Whatman No. 1 paper using a vacuum filter holder set, and then the obtained biomass was washed several times using sterilized deionized water to remove the organic medium [25]. After that, the isolated fungal mycelium was re-suspended with sterilized deionized water to its original volume to form *Mucor* sp. ZG-3 inoculum without PDB, and its concentration was 1.92±0.24 g dry biomass per liter.

*Acidithiobacillus ferrooxidans* LX5 (CGMCC NO.0727) obtained from China General Microbiological Culture Collection Center (CGMCC) was grown in autoclaved modified 9K medium as pure medium [16]. A dosage of 44.2 g/L FeSO$_4$·7H$_2$O as the energy source was added, and the culture pH was then adjusted to 2.5 with dilute sulfuric acid, which is the optimum pH for *A. ferrooxidans* LX5 growth [2, 16]. The inoculum of *A. ferrooxidans* LX5 was prepared in 500 mL Erlenmeyer flasks shaken in a gyratory shaker at 28°C and 180 rpm for 3–4 days until the cell density of *A. ferrooxidans* LX5 reached ~$10^8$ cells/mL [3].

### Evaluation of the effect of *Mucor* sp. ZG-3 on degrading sludge DOM

Municipal sewage sludge was initially autoclaved at 121°C for 15 min, and then primary sludge DOM was obtained by centrifuging autoclaved sludge at 12,500×g for 20 min and then filtering the supernatant through a 0.45 μm membrane [26]. The degradation of primary sludge DOM by *Mucor* sp. ZG-3 was carried out in 250 mL Erlenmeyer flasks containing 90 mL of primary sludge DOM and 10 mL of *Mucor* sp. ZG-3 inoculum (total volume: 100 mL) without PDB (18.1±1.16 mg dry biomass), and the final concentration of sludge DOM in the flasks was 782.4±1.62 mg DOC/L. Control flasks had autoclaved *Mucor* sp. ZG-3 inoculum in place of *Mucor* sp. ZG-3 inoculum. All treatments were done in triplicate. Then, all flasks were shaken in a gyratory shaker at 28°C and 120 rpm for 3 days. During incubation, 10 mL supernatant were withdrawn from flasks daily and filtered through a 0.45 μm membrane to obtain degraded sludge DOM. The concentration of sludge DOM was determined using total organic carbon (TOC) analyzer (Shimadzu TOC-5000A, Japan). Low molecular weight organic acids

(formic acid, acetic acid, propionic acid and butyric acid) in sludge DOM were determined by high-performance liquid chromatography (Waters 600–2487, USA) [16]. Fungal biomass was harvested daily by filtering whole bottle sample through Whatman No. 1 paper, washed with sterilized water, and dried at 80°C for 24 h. The dry weight of fungal biomass was determined to subtract the weight of dried filtering paper from the measured weight. The fungal biomass yield was calculated according to Eq (1):

$$\text{Fungal biomass yield} = \frac{\text{Fungal biomass (dry mass, g)}}{\text{DOM removal (g DOC)}} \tag{1}$$

The loss of water in each flask due to evaporation during incubation was compensated by adding distilled water based on weight loss. Finally, the data presented were the mean values of the triplicate samples with standard deviations.

In order to obtain a more detailed understanding of the role that *Mucor* sp. ZG-3 plays in degrading sludge DOM, the primary sludge DOM and degraded sludge DOM by *Mucor* sp. ZG-3 were grouped into seven fractions of different molecular weights (MW<3000, 3000–4000, 4000–6000, 6000–8000, 8000–10000,1000–14000, and >14000 Da) by using dialysis bags. For each dialysis bag, 5 mL of DOM solution was added and dialyzed against 1 L distilled water in beaker at 4°C. During dialysis, external solution was replaced with distilled water at intervals of 3 h for 12 times over a period of 2 days and the dialysis bag was fully shaken in exchanging process in order to remove the low moleculaer weights fraction. Afterward, each dialyzed sludge DOM was determined by using a TOC analyzer (Shimadzu TOC-5000A, Japan).

## The effect of different molecular weight DOM on Fe$^{2+}$ oxidation by *A. ferrooxidans* LX5

This experiment was firstly carried out to group sludge DOM (primary sludge DOM and degraded sludge DOM by *Mucor* sp. ZG-3) into fractions with different MW, namely, L-DOM (MW<4000 Da), M-DOM (4000 Da<MW<14000 Da), and H-DOM (MW>14000 Da) according to the previous study [27].

500 mL sludge DOM was dialyzed with a dialysis bag of 4000 Da against 500 mL distilled water at 4°C. Dialyzed bag was replaced with new dialysis bag of 4000 Da containing fresh sludge DOM at intervals of 24 h for 3 times over a period of 3 days under the same external solution. After dialysis, the sludge L-DOM was obtained by diluting the external solution to the specified DOM concentration according to the result above.

Sludge DOM was firstly dialyzed with dialysis bag of 4000 Da against distilled water at 4°C, and external solution was replaced with distilled water at intervals of 3 h for 12 times over a period of 2 days to remove the L-DOM fraction. Afterward 500 mL dialyzed sludge DOM without L-DOM was continually dialyzed with dialysis bag of 14000 Da against 500 mL dis-tilled water at 4°C. Dialyzed bag was replaced with new dialysis bag of 14000 Da containing fresh dialyzed sludge DOM without L-DOM at intervals of 24 h for 3 times over a period of 3 days under the same external solution. After dialysis, the sludge M-DOM was obtained by diluting the external solution to the specified DOM concentration according to the result above.

The collection of sludge H-DOM was carried out by dialyzing sludge DOM with dialysis bag of 14000 Da against distilled water at 4°C. The external solution was replaced with distilled water at intervals of 3 h for 12 times over a period of 2 days. After dialysis, the dialyzed sludge DOM (internal solution) was sludge H-DOM.

The inhibition of primary sludge DOM, degraded sludge DOM by *Mucor* sp. ZG-3 and their respective sludge DOM with different MW (L-DOM, M-DOM and H-DOM) to *A. ferrooxidans* LX5 was evaluated in 250 mL Erlenmeyer flasks containing 10 mL, 10 times concentrated modified 9K medium, 28 mL of distilled water, and 60 mL of specified sludge DOM as obtained above [14]. The 2% (v/v) of viable *A. ferrooxidans* LX5 as the inoculum and 10 g/L of FeSO$_4$·7H$_2$O as the energy source for *A. ferrooxidans* LX5 were added into the flasks [16]. Controls were also performed by adding distilled water instead of sludge DOM. The pH of all flasks was adjusted to 2.5 with dilute sulfuric acid, which is the optimum pH for *A. ferrooxidans* LX5 growth [2, 16]. Each treatment was carried out in triplicate, and all flasks were shaken at 28°C and 180 rpm for 6 days [10]. During incubation, 5 mL samples were withdrawn from flasks daily and measured for aqueous Fe$^{2+}$ and Fe$^{3+}$ concentration using the 1, 10-phenanthroline method according to the standard method [22]. The Fe$^{2+}$ oxidation rate was calculated according to Eq (2) [16]:

$$\text{Fe}^{2+} \text{ oxidation efficiency } (\%) = \frac{\text{Initial } (\text{Fe}^{2+}) - \text{Treated } (\text{Fe}^{2+})}{\text{Initial } (\text{Fe}^{2+})} \times 100 \qquad (2)$$

Where (Fe$^{2+}$) is the concentration of Fe$^{2+}$ (mg/L).

## The optimum Fe$^{2+}$ supplementation for sludge dewaterability during the sequential bioleaching process

This experiment was carried out in 1000 mL Erlenmeyer flasks containing 540 mL of municipal sewage sludge and 60 mL of *Mucor* sp. ZG-3 inoculum (1.92±0.24 mg$_{\text{biomass}}$/mL. After one-day treatment of *Mucor* sp. ZG-3, the treated sludge was inoculated with 10% (v/v) of viable *A. ferrooxidans* LX5 inoculum and supplemented with energy source FeSO$_4$·7H$_2$O at different concentrations ranging from 2 to 10 g/L, which is suitable for sludge bioleaching to improve sludge dewaterability. Flow chart of sequential bioleaching process of raw sludge was shown in S1 Fig. Each treatment was carried out in triplicate, and all flasks were shaken at 28°C and 180 rpm for 4 days during the incubation. The treatment without the addition of FeSO$_4$·7H$_2$O was the control treatment. During the treatment period, sludge samples were withdrawn from flasks daily and analyzed for sludge pH, SRF, zeta potential, particle size distribution and concentrations of aqueous Fe$^{2+}$ and Fe$^{3+}$. The SRF was determined using 50 mL of well mixed sludge sample by filtration in a Buchner funnel placed with a filter paper disc (Whatman No. 1) and 0.07 MPa suction pressure [23]. The zeta potential was determined using a Zeta-Plus unit (NanoBrook 90Plus, Brookhaven Instruments, Holtsville, NY). The sludge particle size distribution was measured by laser scattering image analysis (Mastersizer2000, Malvern Panalytical). The morphology and characteristics of raw sludge sample and treated sludge were analyzed using scanning electron microscopy (SEM, SU8600, JEOL, Japan).

## Evaluation of sequential and conventional bioleaching process on sludge dewaterability under optimum Fe$^{2+}$ supplementation

This experiment was carried out in 1000 mL Erlenmeyer flasks. The sequential bioleaching process under optimum Fe$^{2+}$ supplementation was carried out according to the result above. The treatments with the addition of optimum Fe$^{2+}$ supplementation and 10 g/L energy source (optimum Fe$^{2+}$ supplementation for conventional bioleaching process) and without energy source in conventional bioleaching process were also performed. Each treatment was carried out in triplicate, and all flasks were shaken at 28°C and 180 rpm for 4 days during the

incubation. During the treatment period, 50 mL sludge samples were withdrawn from flasks daily for sludge SRF measurement.

All measures are listed in S1 Dataset.

## Results and discussion

### Effect of *Mucor* sp. ZG-3 on degrading sludge DOM

Sewage sludge DOM harbors a plethora of organic compounds that serve as carbon and nutrient sources for heterotrophic organisms. Certain filamentous fungi possess the capability to transform or break down organic constituents present in sludge DOM [19, 25, 28–30]. As depicted in Fig 1, the concentration of primary sludge DOM, inoculated with *Mucor* sp. ZG-3, notably decreased from an initial 782 mg DOC/L to 412 mg DOC/L after a day of fungal treatment, indicating the effective degradation of sludge DOM by *Mucor* sp. ZG-3. Meanwhile, the fungal biomass and yield reached 1.12 g dry weight/L and 2.53 g dry weight/g DOC, respectively. Notably, the conversion of sludge DOM primarily contributed to the biomass of *Mucor* sp. ZG-3 itself. Throughout the incubation period, the mycelium exhibited a filamentous and evenly dispersed morphology, facilitating optimal contact between the mycelium and sludge DOM, thereby enhancing DOM degradation efficiency. Indeed, filamentous fungi can augment their biomass by redirecting their secondary metabolism to assimilate soluble substances

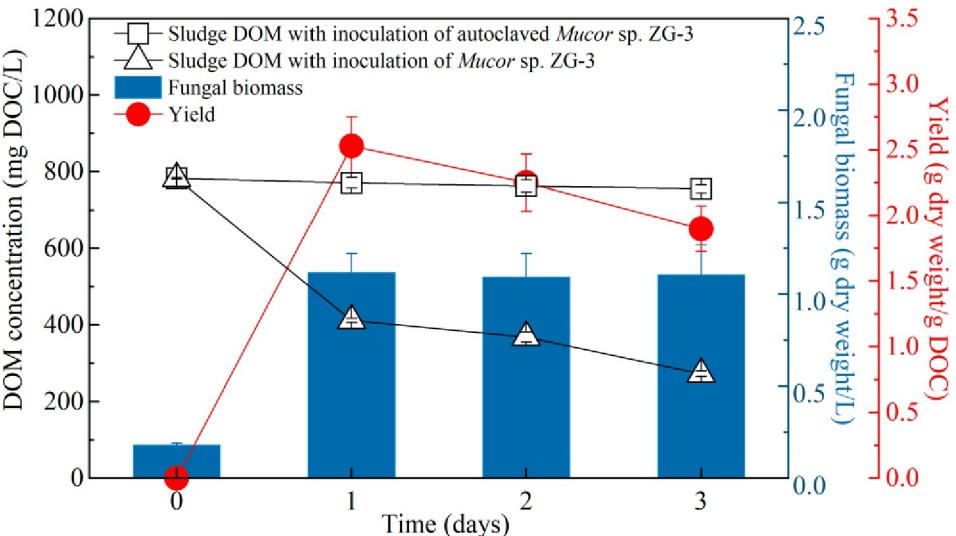

**Fig 1. Dynamics of sludge DOM, fungal biomass and yield with treatment time in primary sludge DOM with inoculation of *Mucor* sp. ZG-3 and without any inoculation.** Sludge DOM comprises a heterogeneous mixture of diverse organic compounds with varying molecular weights. As illustrated in Fig 2, the primary sludge DOM in this study primarily consisted of DOM with MW <4000 Da and >14000 Da, accounting for 75.7% and 22.6%, respectively. The remaining sludge DOM content (4000 Da<MW<14000 Da) was negligible. A previous study grouped sludge DOM into fractions based on MW, namely, L-DOM (MW<4000 Da), M-DOM (4000 Da<MW<14000 Da), and H-DOM (MW>14000 Da) [27]. After 3 days of fungal treatment, the concentration of DOM with MW<3000 Da sharply declined from an initial value of 549.8 mg DOC/L to 124.6 mg DOC/L, while DOM with 3000 Da<MW<4000 Da degraded from an initial 22.2 mg DOC/L to 11.6 mg DOC/L, indicating a 76.2% of reduction in L-DOM eventually. Moreover, *Mucor* sp. ZG-3 also contributed to a 27.4% degradation of H-DOM, resulting in a concentration of 124.0 mg DOC/L in sludge DOM. These findings indicate that *Mucor* sp. ZG-3 not only assimilates L-DOM significantly (76.2% reduction) but also partly degrades H-DOM (27.4% reduction). These observations can be primarily attributed to the heterotrophic nature of *Mucor* sp. ZG-3, which can rapidly assimilate nutritional substances in L-DOM to form biomass [14]. Additionally, the secretion of certain enzymes by filamentous fungi facilitates the breakdown or transformation of H-DOM into degradable L-DOM [28, 31, 32].

from sludge, a phenomenon consistent with previous findings [19, 28, 31–33]. As the metabolism of DOM progressed and nutritional sources were depleted by *Mucor* sp. ZG-3, fungal biomass ceased to increase (maintaining at 1.11 g dry weight/L), while DOM concentration and yield continued to decline, reaching 272 mg DOC/L (a degradation of 65.2%) and 1.90 g dry weight/g DOC by day 3, respectively. Similar outcomes were reported by [34], where the secretion of several enzymes by high microbial growth facilitated the substantial degradation of organic substrates in sludge. However, the concentration of primary sludge DOM inoculated with autoclaved *Mucor* sp. ZG-3 remained relatively unchanged during incubation, likely because of the sorption of fungal mycelium. Consequently, the degradation performance of autoclaved *Mucor* sp. ZG-3 on sludge DOM was minimal. Thus, the efficacy of *Mucor* sp. ZG-3 in sludge DOM degradation can be attributed to assimilation and biodegradation rather than organic substance sorption.

## Effect of *Mucor* sp. ZG-3 on alleviating the inhibition of sludge DOM to *A. ferrooxidans* LX5

As depicted in Fig 2, the primary sludge DOM mainly consisted of L-DOM and H-DOM. Consequently, the inhibition of sludge DOM on *Acidithiobacillus* species can be attributed to the presence of L-DOM or H-DOM. As illustrated in Fig 3, the oxidation of $Fe^{2+}$ did not occur within 6 days of inoculation in the pure medium supplemented with primary L-DOM, whereas $Fe^{2+}$ oxidization efficiency reached 100% within only 2 days in the pure media with H-DOM, a trend consistent with that observed in the modified 9K medium. Hence, L-DOM, unlike H-DOM, tends to inhibit $Fe^{2+}$ oxidation by *Acidithiobacillus* species. Studies have identified certain low-molecular-weight organic acids as severe inhibitors of the activities and growth of

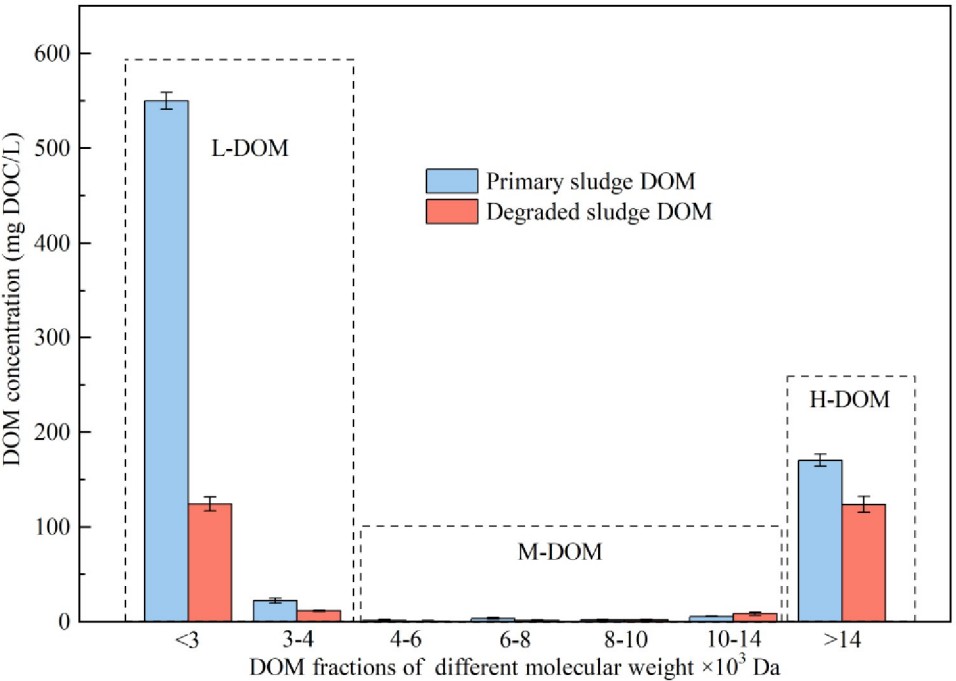

**Fig 2. DOM concentration of different molecular weight fractions in primary sludge DOM and degraded sludge DOM by *Mucor* sp. ZG-3.** Primary sludge DOM mainly consisted of L-DOM and H-DOM. *Mucor* sp. ZG-3 not only assimilates L-DOM significantly but also partly degrades H-DOM.

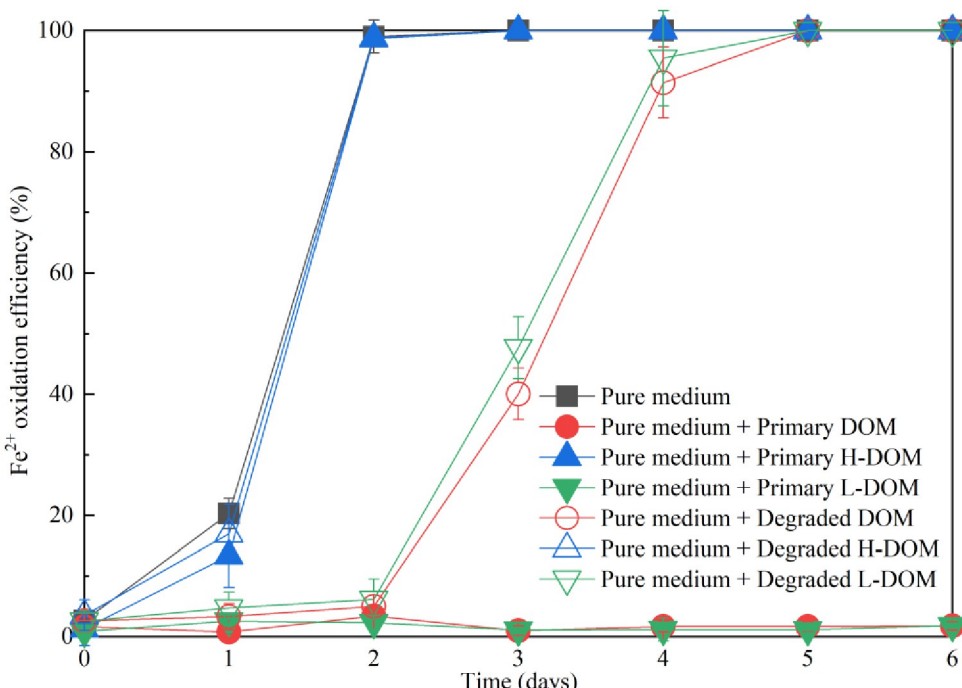

**Fig 3. Change of Fe$^{2+}$ oxidation efficiency achieved by *A. ferrooxidans* LX5 grown in different medium.** Primary L-DOM is toxic to *A. ferrooxidans* LX5, while primary H-DOM is as safe as pure medium to *A. ferrooxidans* LX5. *Mucor* sp. ZG-3 could alleviate sludge DOM inhibition to *A. ferrooxidans* LX5.

*Acidithiobacillus* species in sludge DOM [16, 35, 36]. These low-molecular-weight organic acids are predominantly present in L-DOM. Specific low-molecular-weight organic acids with significant inhibitory effects on Fe$^{2+}$ oxidation by *Acidithiobacillus* species in sludge DOM were also detected, as depicted in Fig 4. Formic acid, acetic acid, and butyric acid were completely depleted after 3 days of fungal treatment, while the content of propionic acid declined from 194.6 mg/L to 92.3 mg/L (a degradation of 52.6%). With the depletion and degradation of these significant inhibitory low-molecular-weight organic acids, the degraded sludge DOM by *Mucor* sp. ZG-3 becomes more conducive to the growth of *Acidithiobacillus* species. Similar results were reported by [37], where an increase in *Acidithiobacillus* species was accompanied by the consumption of acetic acid and propionic acid during sludge bioleaching. In the present study, the fungal degradation efficiency of L-DOM, which exhibits severe inhibition toward *Acidithiobacillus* species, reached 76.2%. Consequently, the inoculation of *Mucor* sp. ZG-3 alleviates the inhibition of sludge DOM on *A. ferrooxidans* LX5 due to the extensive degradation of L-DOM by *Mucor* sp. ZG-3, as also validated in Fig 3. The trend in Fe$^{2+}$ oxidation was essentially the same in the pure medium supplemented with both primary sludge DOM and L-DOM. Fe$^{2+}$ oxidation did not occur within 6 days of inoculation in the pure medium supplemented with primary sludge DOM. The result was lower than nearly 20% Fe$^{2+}$ oxidation efficiency reported in a previous study [14]. This difference can be attributed to two factors: the higher concentration of DOM in our study compared with that reported previously and the possibility of certain substances in sludge causing rapid Fe$^{2+}$ consumption due to chemical reactions in the early stages because variations in sampling time can affect sludge properties. A similar trend was observed in the L-DOM medium, indicating the severe inhibition of both primary sludge DOM and L-DOM on the activity of *A. ferrooxidans*

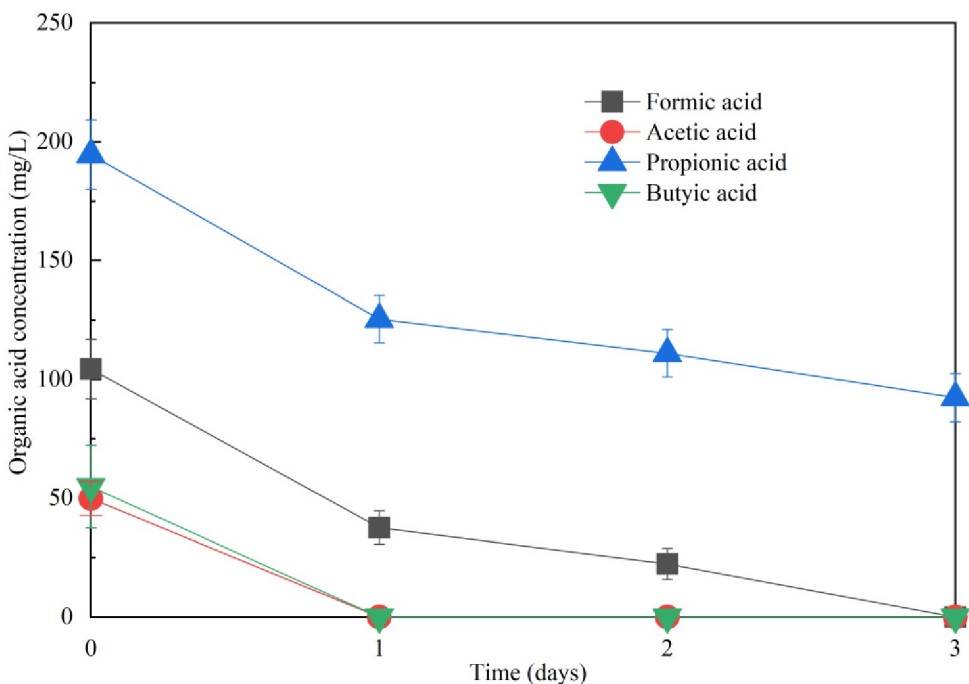

**Fig 4. Variation of low molecular weight organic acids in degraded sludge DOM by *Mucor* sp. ZG-3 during incubation.**

LX5. However, the inhibitory effect was markedly mitigated after *Mucor* sp. ZG-3 degraded primary sludge DOM, with nearly 90% of $Fe^{2+}$ in the pure medium supplemented with degraded sludge DOM oxidized within only 4 days. Consequently, the inhibition of sludge DOM on *A. ferrooxidans* LX5 was significantly alleviated. In conclusion, *Mucor* sp. ZG-3, similar to conventional sludge DOM degraders such as *R. mucilaginosa* R30 and *P. spartinae* D13, plays a crucial role in alleviating sludge DOM inhibition on *A. ferrooxidans* LX5 and enhancing the activity of *A. ferrooxidans* LX5 [10, 11, 14].

## Effect of $Fe^{2+}$ supplementation on sludge dewaterability during sequential bioleaching process

Compared with the conventional sludge bioleaching process, the sequential sludge bioleaching process substantially degrades sludge DOM, providing a more favorable environment for the growth of *A. ferrooxidans* LX5. This, in turn, may reduce the demand for energy sources to enhance sludge dewaterability. The effect of $Fe^{2+}$ supplementation on sludge dewatering performance under the sequential inoculation process was investigated, and the results are presented in Fig 5. The SRF decreased sharply from $2.63 \times 10^{12}$ m/kg to $0.54$–$0.60 \times 10^{12}$ m/kg after the inoculation of *A. ferrooxidans* LX5 culture at the end of day 1. The inoculation of *A. ferrooxidans* LX5 into sludge treated by *Mucor* sp. ZG-3 led to a significant reduction in SRF, consistent with observations reported in previous study indicating the positive effect of *A. ferrooxidans* LX5 culture on dewaterability [14]. The negligible difference in SRF among treatments with varying amounts of $FeSO_4 \cdot 7H_2O$ supplementation at the inoculation point (day 1) suggests that sludge dewaterability was primarily enhanced chemically by the *A. ferrooxidans* LX5 culture rather than the energy source. This finding aligns with previous studies indicating that *A. ferrooxidans* LX5 culture is more effective than energy sources in enhancing sludge

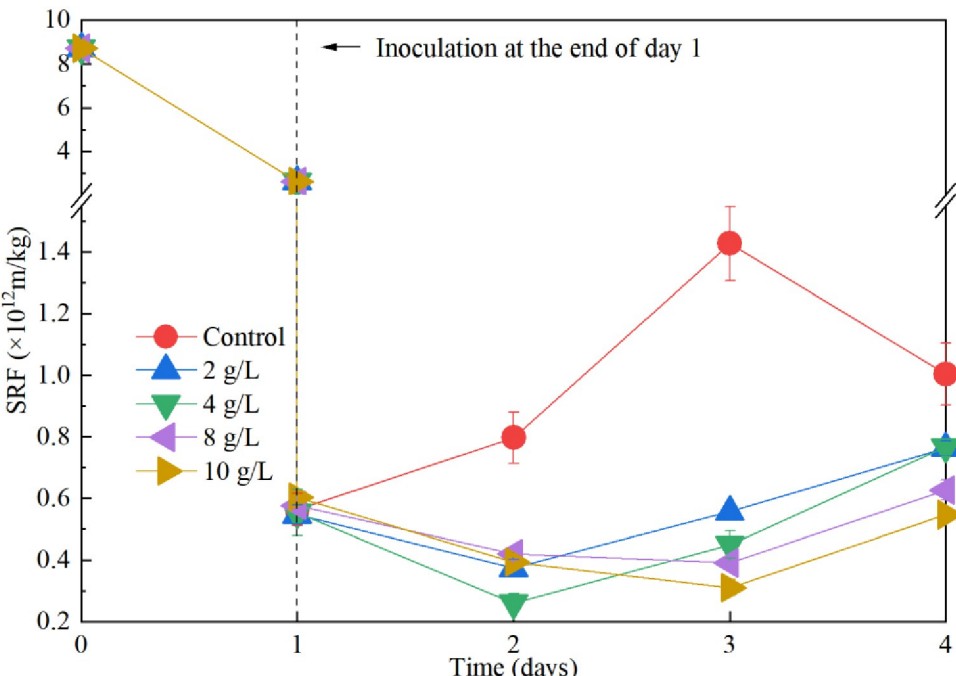

**Fig 5. Change of sludge SRF with treatment time in sequential bioleaching process with different energy source supplementation.** Treatment with addition of 4 g/L energy source achieved the lowest SRF value after two days of incubation.

dewaterability initially [2, 4, 5]. The absence of energy source cannot support the growth of *A. ferrooxidans* LX5, leading to sludge deterioration and a gradual increase in SRF to its highest value of $1.43 \times 10^{12}$ m/kg by day 3, whereas SRF continues to decrease in the presence of energy source. The SRF values of treatments with 2 g/L and 4 g/L energy source additions were reduced to their respective lowest values after 1 day treatment with *A. ferrooxidans* LX5, while achieving the lowest SRF values through treatments with 8 g/L and 10 g/L energy source additions required 2 days. This delay can be attributed to the additional time required for *A. ferrooxidans* LX5 to consume excess energy sources. 4 g/L energy source was optimal for sludge dewaterability in the sequential inoculation process, with SRF minimized to the lowest value of $2.60 \times 10^{11}$ m/kg, representing a 96.9% reduction. However, sludge dewatering performance deteriorated with increasing $Fe^{2+}$ concentration after energy source supplementation beyond 4 g/L. Wong et al. concluded that sludge dewaterability deteriorated during conventional bioleaching when the $Fe^{2+}$ concentration exceeded 2 g/L, equivalent to 10 g/L of energy source [12]. Although SRF values in treatments with 8 g/L and 10 g/L energy source additions fluctuated during incubation, their lowest SRF values remained higher than those achieved with 4 g/L energy source addition. The lowest values of sludge SRF and corresponding treatment times under sequential inoculation with different energy source additions over a 4-day treatment period are summarized in Table 1. Notably, only treatment with of 4 g/L energy source addition achieved the lowest SRF value of $2.60 \times 10^{11}$ m/kg after 2 days of treatment, representing a 30.1%, 33.3%, and 16.4% reduction compared with treatments with 2 g/L, 8 g/L, and 10 g/L energy source additions, respectively. Therefore, compared with the conventional bioleaching process, the sequential bioleaching process not only reduces the demand for energy sources for *A. ferrooxidans* LX5 but also achieves better sludge dewaterability performance in less time.

**Table 1. Lowest sludge SRF achieved and time needed under sequential bioleaching process with different energy source addition.**

| Energy source addition (g/L) | Lowest SRF ($\times 10^{11}$ m/kg) | Time needed |
|:---:|:---:|:---:|
| 0 (control) | 5.63($\pm$0.51) | 1 day |
| 2 | 3.73($\pm$0.29) | 2 days |
| 4 | 2.60($\pm$0.30) | 2 days |
| 8 | 3.90($\pm$0.20) | 3 days |
| 10 | 3.11($\pm$0.26) | 3 days |

Notes, SRF: specific resistance to filtration.

## Effect of $Fe^{2+}$ supplementation on sludge pH, surface charge, $Fe^{2+}$ oxidation, and floc size during sequential bioleaching process

Fig 6 illustrates the variations in pH and surface charge of the sludge during incubation. As depicted in Fig 6A, inoculating *A. ferrooxidans* LX5 into the treated sludge by *Mucor* sp. ZG-3, at the end of day 1, following the addition of energy source, resulted in a decrease in sludge pH from 5.83 to 4.23–4.42. The negligible difference in pH values among treatments with varying energy source additions suggests that the pH decrease due to acidification is attributable to the *A. ferrooxidans* LX5 culture, rather than the energy source. Huo et al. observed similar results: *A. ferrooxidans* LX5 culture was more effective than the energy source in reducing sludge pH initially [2]. The absence of energy source in sludge cannot sustain the growth of *A. ferrooxidans* LX5, leading to a gradual increase in sludge pH [12]. Thus, energy source is crucial for the bio-acidification of sludge by *A. ferrooxidans* LX5, as confirmed in other treatments in the presence of energy source. The hydrolysis of $Fe^{3+}$ oxidized from $Fe^{2+}$ by *A. ferrooxidans* LX5 releases $H^+$, which is responsible for the pH reduction, and the extent of pH drop directly correlates with the amount of $Fe^{3+}$ [12, 35]. Sludge acidification can disrupt flocs, releasing bound water, and neutralize sludge particles, making them more easily aggregated, thereby enhancing sludge dewaterability [38–40]. Additionally, surface charge is a crucial factor affecting sludge dewatering. As depicted in Fig 6B, the trend in zeta potential change was consistent with that of pH. With increasing energy source supplementation, the sludge zeta potential increased. Particularly, the sludge zeta potential increased from -31.8 mV in raw sludge to -9.47 to -8.23 mV within the first 2 days when the energy source concentration exceeded 4 g/L, indicating a significant decrease in the net surface charge on the flocs. As reported in the literature, sludge flocs with minimal or absent surface charge tend to aggregate easily and remain firmly bound, resulting in favorable sludge dewatering performance. Liu et al. observed improved sludge dewaterability in bio-acidified sludge, attributed to a zeta potential close to 0 mV when the pH was approximately 2.4 [3].

However, the pH and zeta potential ceased to change significantly when the energy source concentration exceeded 4 g/L by the end of day 2. The stabilization might be attributed to the assumption that the oxidation rate of $Fe^{2+}$ by *A. ferrooxidans* LX5 had already reached its peak on the first day. Indeed, the concentration of $Fe^{3+}$ oxidized from $Fe^{2+}$ and the oxidation efficiency of $Fe^{2+}$, as depicted in Fig 7, confirmed this speculation. The concentration of $Fe^{3+}$ oxidized from $Fe^{2+}$ by *A. ferrooxidans* LX5 in Fig 7A was obviously increased with the increase of $Fe^{2+}$ supplementation. The concentration of $Fe^{3+}$ in treatments with 2 g/L and 4 g/L additions was 452.8 mg/L and 735.4 mg/L, respectively. However, no discernible changes were observed when the energy source addition exceeded 4 g/L by the end of day 2. The oxidation efficiency of $Fe^{2+}$, as illustrated in Fig 7B, indicated the effective bio-oxidation of $Fe^{2+}$ by *A. ferrooxidans*

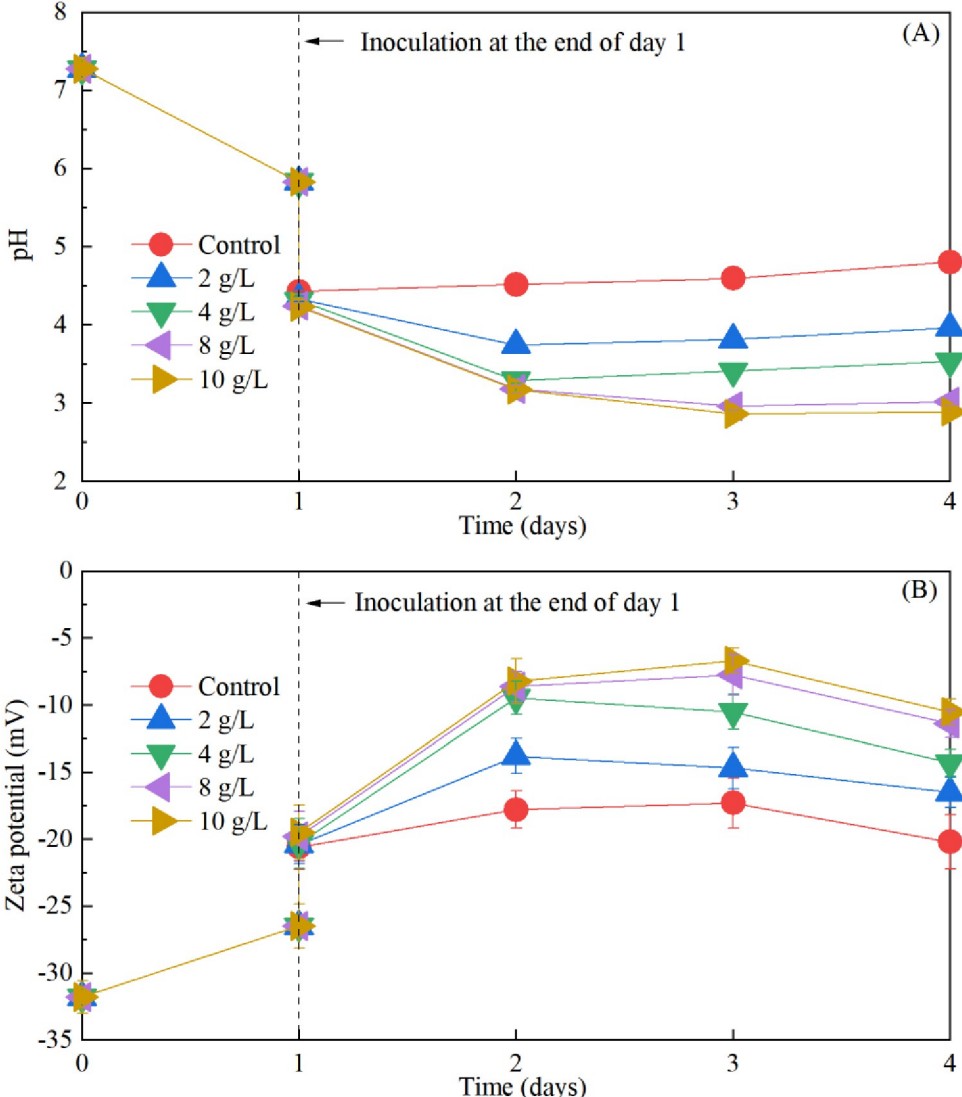

**Fig 6. Changes of sludge pH (A) and zeta potential (B) with treatment time in sequential sludge bioleaching process with different energy source supplementation.**

LX5, resulting in $Fe^{3+}$ production that facilitated sludge flocculation and improved dewaterability. Treatment with 2 g/L energy source achieved 100% oxidation efficiency of $Fe^{2+}$, but the $Fe^{3+}$ concentration was only 452.8 mg/L. Conversely, treatment with 4 g/L energy source reached a 90% oxidation efficiency of $Fe^{2+}$ and a $Fe^{3+}$ concentration of 735.4 mg/L. Hence, it was the addition of 4 g/L energy source, rather than 2 g/L, that sufficiently supported the growth of *A. ferrooxidans* LX5 and optimized sludge bio-acidification and dewaterability after treatment for 1 day.

Furthermore, after sequential bioleaching treatment for 2 days, the particle size of the sludge is depicted in Fig 8. A notable increase in the median particle size (d50) of the sludge flocs occurred, rising from 17.90 μm of raw sludge to 21.87–34.32 μm. Fungal mycelia entrapped sludge particles, leading to the formation of larger sludge flocs, which enhanced the strength and rigidity of the fungal-treated sludge [33]. This entanglement with fungal mycelia

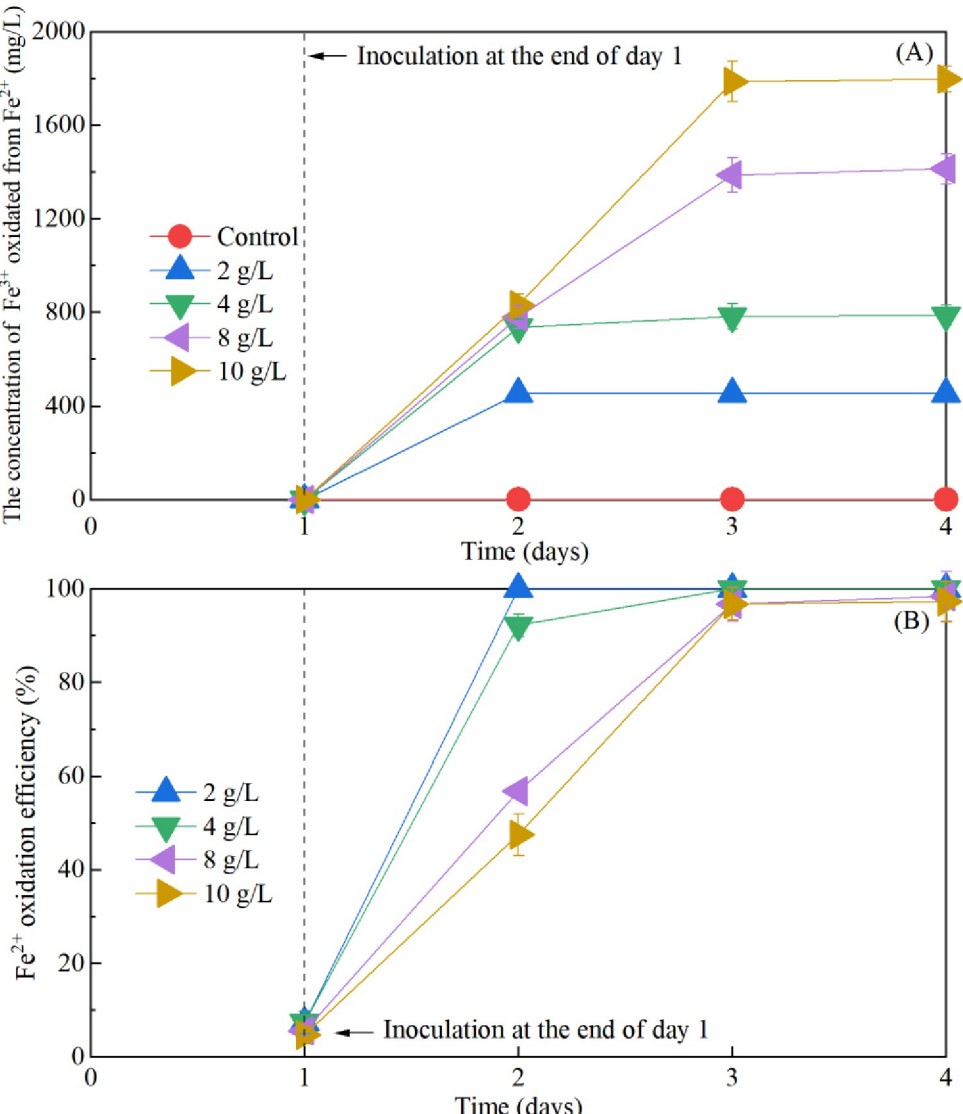

**Fig 7. Changes of the concentration of $Fe^{3+}$ oxidized from $Fe^{2+}$ (A), and $Fe^{2+}$ oxidation efficiency (B) with treatment time in sequential sludge bioleaching process with different energy source supplementation.**

enabled the sludge to maintain high permeability during pressure filtration and created spaces for the outflow of free water [33]. However, with the increase in energy source supplementation, the d50 of the sludge particles gradually decreased due to the breakdown of sludge flocs through bio-acidification, releasing trapped water within the flocs. When the energy source supplementation exceeded 4 g/L, the d50 of the sludge flocs decreased to 21.87–22.48 μm, potentially leading to blockage of the dewatering tunnel within the fungal-treated sludge, thereby deteriorating sludge dewatering performance. Scanning electron microscopy (SEM) images of raw sludge and treated sludge exhibiting optimal dewaterability performance are presented in Fig 9. In comparison to the dispersed small sludge particles in raw sludge (Fig 9A), sludge particles in the sequential bioleaching process (Fig 9B) were trapped by fungal mycelia, resulting in larger sludge particles that enhanced sludge dewaterability [25, 41, 42].

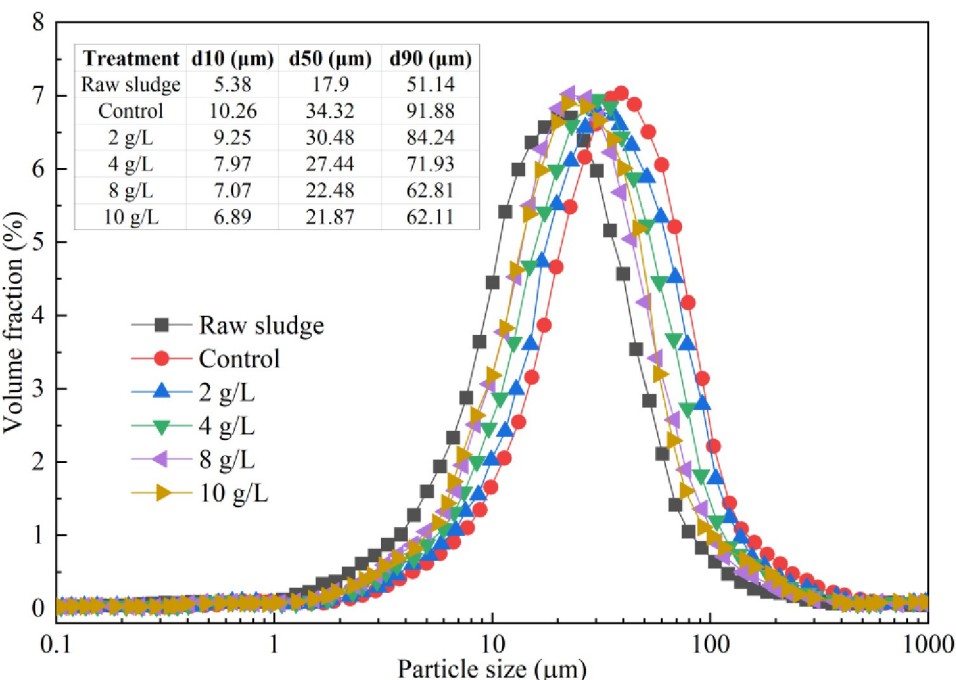

| Treatment | d10 (µm) | d50 (µm) | d90 (µm) |
|---|---|---|---|
| Raw sludge | 5.38 | 17.9 | 51.14 |
| Control | 10.26 | 34.32 | 91.88 |
| 2 g/L | 9.25 | 30.48 | 84.24 |
| 4 g/L | 7.97 | 27.44 | 71.93 |
| 8 g/L | 7.07 | 22.48 | 62.81 |
| 10 g/L | 6.89 | 21.87 | 62.11 |

**Fig 8. Changes in particle size distribution of raw sludge and treated sludge in sequential bioleaching process for two days with different energy source supplementation.**

## Evaluation of sequential and conventional bioleaching process on sludge dewaterability under respective optimum $Fe^{2+}$ supplementation

As depicted in Fig 10, sludge dewaterability gradually improved with increased energy source supplementation in the conventional bioleaching process. This finding is consistent with numerous studies [3, 12, 21]. The sludge SRF value minimized to the lowest value of $4.33\times10^{11}$ m/kg with an optimum energy source supplementation of 10 g/L in the conventional bioleaching process [12], which, however, remained higher than that in the sequential bioleaching

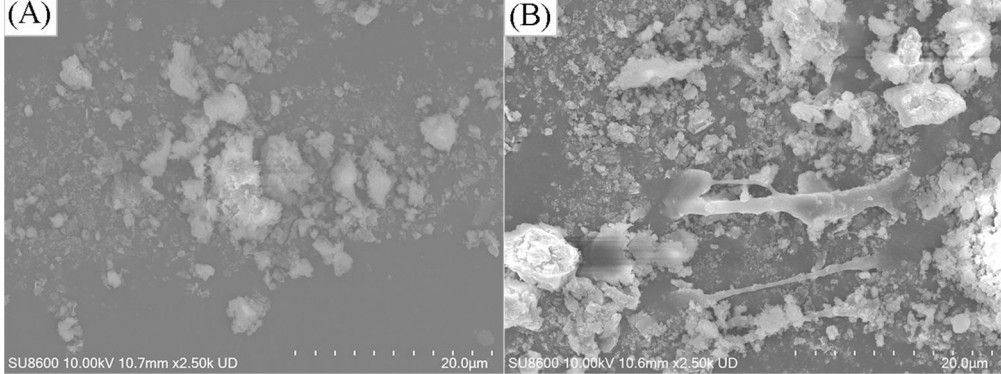

**Fig 9. SEM images of raw sludge (A) and sequential bioleaching process with the addition of 4 g/L energy source for two days (B).**

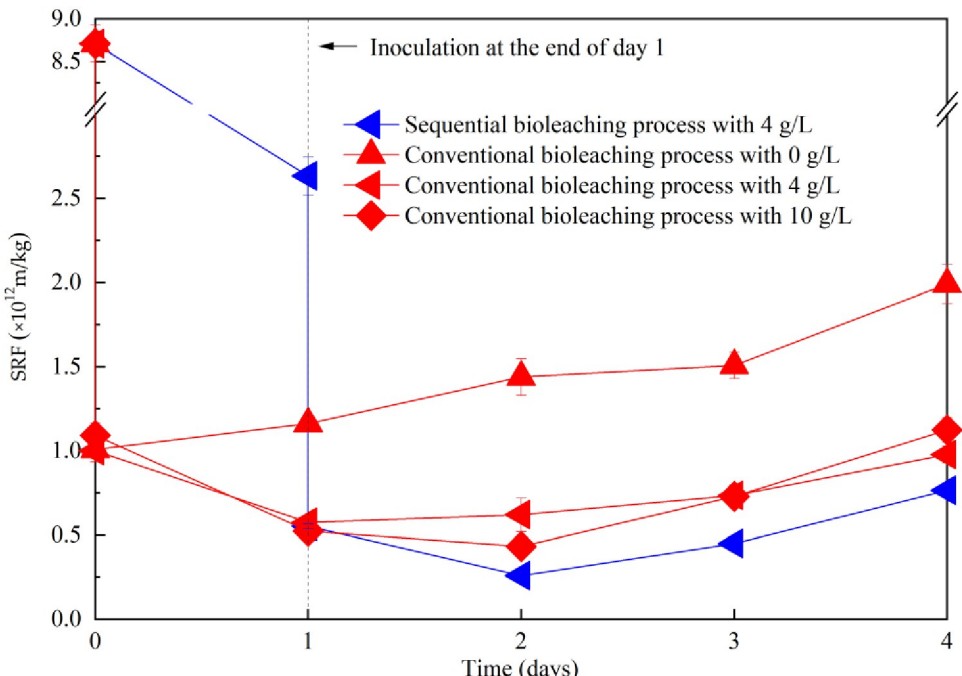

**Fig 10. Change of sludge SRF with treatment time in municipal sewage sludge with sequential bioleaching process with the addition of energy source at 4 g/L, and conventional bioleaching process with the addition of energy source at 0 g/L, 4 g/L and 10 g/L.**

process with the addition of 4 g/L energy source. The lowest sludge SRF value in the sequential bioleaching process was still 54.9% and 40.0% lower than that in the conventional bioleaching process with the addition of 4 g/L and 10 g/L energy sources, respectively. Thus, the sequential bioleaching process not only enhanced sludge dewaterability performance but also reduced the demand for energy sources. Compared with traditional bioleaching process, the sludge cake obtained through mechanical dewatering of sludge treated in the sequential inoculation process would contain low water content and a high proportion of volatile suspended solids. This can be attributed to the high amount of filamentous fungal biomass and low content of inorganic biogenic flocculant present in the sludge cake, facilitating composting and incinerating [43, 44].

## Conclusions

*Mucor* sp. ZG-3 demonstrated effectiveness in degrading sludge DOM by partly degrading H-DOM, which presented no inhibition to *A. ferrooxidans* LX5, and substantially assimilating L-DOM, which was detrimental to *A. ferrooxidans* LX5. The fungus efficiently degraded low-molecular-weight organic acids within L-DOM in the sludge DOM, notably alleviating sludge DOM inhibition on *A. ferrooxidans* LX5. Consequently, this created a more favorable environment for the growth of *A. ferrooxidans* LX5. In the sequential bioleaching process, the optimal addition of $FeSO_4 \cdot 7H_2O$ was determined to be 4 g/L. This resulted in a substantial increase in zeta potential and particle size, contributing to improved sludge dewaterability. However, a higher concentration of $FeSO_4 \cdot 7H_2O$ did not yield further enhancements in sludge dewaterability. Therefore, the sequential bioleaching process not only reduced the demand for energy sources for *A. ferrooxidans* LX5 but also achieved superior sludge dewaterability performance.

Therefore, future studies must explore whether a consecutive multibatch bioleaching process involving the circulation of bioleached sludge could achieve optimal dewatering performance.

## Supporting information

**S1 Fig. Flow chart of sequential bioleaching process of sewage sludge.**
(DOCX)

**S1 Dataset.**
(XLSX)

## Acknowledgments

We are grateful to the Solid Waste Institute of Nanjing Agricultural University on providing relevant microorganisms.

## Author Contributions

**Conceptualization:** Zhenyu Wang, Guanyu Zheng, Lixiang Zhou.

**Funding acquisition:** Zhenyu Wang.

**Investigation:** Zhenyu Wang, Wen Feng, Shigang Tang.

**Methodology:** Zhenyu Wang, Wen Feng, Shigang Tang, Jing Zhao.

**Project administration:** Zhenyu Wang.

**Resources:** Zhenyu Wang, Wen Feng, Shigang Tang, Guanyu Zheng, Lixiang Zhou.

**Supervision:** Zhenyu Wang.

**Validation:** Zhenyu Wang, Shigang Tang.

**Visualization:** Zhenyu Wang, Wen Feng, Jing Zhao.

**Writing – original draft:** Zhenyu Wang, Jing Zhao.

**Writing – review & editing:** Zhenyu Wang.

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
