## [Decision Letter · Decision Letter 0]

27 Feb 2024

PONE-D-24-03890Filamentous fungus Mucor sp. ZG-3 degrades inhibitory substances and improves sludge dewaterability during sequential bioleaching with Acidothiobacillus ferrooxidans LX5PLOS ONE

Dear Dr. Wang,

Thank you for submitting your manuscript to PLOS ONE. After careful consideration, we feel that it has merit but does not fully meet PLOS ONE’s publication criteria as it currently stands. Therefore, we invite you to submit a revised version of the manuscript that addresses the points raised during the review process.

We look forward to receiving your revised manuscript.

Kind regards,

Rakesh Namdeti

Academic Editor

PLOS ONE

Journal Requirements:

"This research was funded by Basic Public Welfare Research Program of Zhejiang Province ZW (LGF21B070001), Lishui Public Welfare Technology Application Research Project ZW (2021GYX13), Lishui Public Welfare Technology Application Research Project ST (2022GYX05), and National Natural Science Foundation of China ZW(21707060)."

Please state what role the funders took in the study.  If the funders had no role, please state: ""The funders had no role in study design, data collection and analysis, decision to publish, or preparation of the manuscript."" If this statement is not correct you must amend it as needed. 

**Additional Editor Comments:**

- Improve clarity and coherence of language to enhance reader comprehension.

- Ensure adherence to the specific formatting and submission guidelines outlined by the journal.

- Provide detailed explanations of methodology and analysis techniques for better transparency.

- Engage with existing literature more comprehensively to situate your research within the broader context of the field.

- Address any reviewer feedback or suggestions in a thorough and timely manner during the revision process.

Reviewers' comments:

Reviewer's Responses to Questions

**Comments to the Author**

1. Is the manuscript technically sound, and do the data support the conclusions?

Reviewer #1: Yes

Reviewer #2: Yes

Reviewer #3: Yes

2. Has the statistical analysis been performed appropriately and rigorously? 

Reviewer #1: Yes

Reviewer #2: Yes

Reviewer #3: Yes

3. Have the authors made all data underlying the findings in their manuscript fully available?

Reviewer #1: Yes

Reviewer #2: Yes

Reviewer #3: Yes

4. Is the manuscript presented in an intelligible fashion and written in standard English?

Reviewer #1: No

Reviewer #2: Yes

Reviewer #3: Yes

5. Review Comments to the Author

Reviewer #1: The manuscript by Wang et al. described the role of Filamentous fungus Mucor sp. ZG-3 in eliminating inhibitory substances to Acidithiobacillus ferrooxidans LX5 and improving sludge dewaterability by bioleaching. The authors found that Mucor sp. ZG-3 could effectively degrade Low-molecular-weight dissolved organic matter toxic to A.f and improved sludge dewaterability. Moreover, the energy substance amount required in bioleaching with sequential inoculation Mucor sp.ZG-3 and A.f LX5 could be drastically reduced to 4 g/L, which would decrease operation cost in engineering application. Generally speaking, the manuscript is interesting and have some new information on how to enhance bioleaching effectiveness. However, there are many concerns arisen in the manuscript. I recommended to make a major revision prior to accepting it for publication.

1. The topic of the manuscript should be rephrased since the present topic is very similar to the published paper (Chemical Eng J, 2016,284:216-223) although their content is different.

2. L20-21, “The sludge DOM decreased to 272 mg dissolved organic carbon (DOC)/L with 65.2% reduction by Mucor sp. ZG-3 in 3 days” changed into “Sludge dissolved organic carbon (DOC) decreased to 272 mg /L with 65.2% reduction by Mucor sp. ZG-3 in 3 days”

3. L77-78, “390.85 mg dissolved organic carbon (DOC)/L” changed into “390.85 mg DOC/L”

4. L32, what means is d50 ? You should provide the full name of abbreviation when it appeared in the first time.

5. L39, “findings”�”studies”

6. L102-104, “…purposes: (1) investigated……; (2) evaluate……”�“…purposes to (1) investigate……; and (2) evaluate…….

7. L116, “specific resistance to filtration (SRF)” �“SRF“

8. L118 and Table 1, delete “The solid sludge content”. I suggest to delete Table 1. The data in Table 1 can insert into the text.

9. L183, please describes dialysis process in details including how many volumes of distilled water dialysis bag is placed into. Exchange times of distilled water? Does it need to shake or stir during dialysis?

10. Subtitle “Evaluation of the effect of sludge DOM with different MW on Fe2+ oxidation by A. ferrooxidans LX5” cannot stand for the content of the text. Please rephrase it. It seems to be that “The effect of different molecular weight DOM on Fe2+ oxidation by A.ferrooxidans LX5”.

11. L194-212. What is DOM concentration obtained by autoclaving, centrifuging and filtering sludge? How do you get 572 mgDOC/L of sludge L-DOM? Doesn’t it need to freeze-drying dialyzed external solution and then re-dissolve it to obtain a given concentration of L-DOM? Likewise, what about other fractions of DOM with different MW? Please check it carefully and describe the procedures.

12. L232, delete “Evaluation of”

13. There are too many grammar and syntax errors in the manuscript. Besides, Table and figures are also irregular.

Reviewer #2: The results were nicely presented. The following corrections needs to be addressed.

Page 12: equation 2 – mention the actual formula, whcih can make readers easy to understand. (not the values in formula)

Page 19: Line 381 – the statement “It is L-DOM not H-DOM….” is little confusing. Better the statement can be rephrased for better understanding.

Fig 8 – Y axis – volume fraction cannot have %. It is a fraction.

Fig 6 – Inoculation at the end of day 1 – the vertical line is shown not at 1 day in X axis – Error needs to be corrected in both graph 6A and 6B.

Reviewer #3: Provide keywords at the end of abstract.

Research gap is essential between literature review and objectives. It is missing in the manuscript.

Novelty statement is not clear.

Background, methods, results, discussion and conclusion are written very well.

References section is not updated. Only 7 out of 44 is after 2020.

Authors may provide supplementary information within in the manuscript.

6. PLOS authors have the option to publish the peer review history of their article (what does this mean?). If published, this will include your full peer review and any attached files.

Reviewer #1: No

Reviewer #2: No

Reviewer #3: **Yes,** Dr. S. Sivamani

---

## [Author Response · Author response to Decision Letter 0]

20 Mar 2024

Response to the Editor Comments

Additional Editor Comments:

1. Improve clarity and coherence of language to enhance reader comprehension.

Reply: Thank you very much for your suggestion. Language has been refined and revised throughout the text in the revised manuscript.

2. Ensure adherence to the specific formatting and submission guidelines outlined by the journal.

Reply: Thank you very much for your suggestion. This paper strictly follows the specific formatting and submission guidelines of the journal.

3. Provide detailed explanations of methodology and analysis techniques for better transparency.

Reply: Thank you very much for your suggestion. The detailed explanations of methodology and analysis techniques have been provided in the revised manuscript.

4. Engage with existing literature more comprehensively to situate your research within the broader context of the field.

Reply: Thank you very much for your suggestion. The abstract has been rewritten in the revised manuscript as instructed.

5. Address any reviewer feedback or suggestions in a thorough and timely manner during the revision process.

Reply: Thank you very much for your suggestion. All feedback or suggestions from any reviewers have been addressed thoroughly and timely.

Response to the Reviewer #1:

Reviewer #1: The manuscript by Wang et al. described the role of Filamentous fungus Mucor sp. ZG-3 in eliminating inhibitory substances to Acidithiobacillus ferrooxidans LX5 and improving sludge dewaterability by bioleaching. The authors found that Mucor sp. ZG-3 could effectively degrade Low-molecular-weight dissolved organic matter toxic to A.f and improved sludge dewaterability. Moreover, the energy substance amount required in bioleaching with sequential inoculation Mucor sp.ZG-3 and A.f LX5 could be drastically reduced to 4 g/L, which would decrease operation cost in engineering application. Generally speaking, the manuscript is interesting and have some new information on how to enhance bioleaching effectiveness. However, there are many concerns arisen in the manuscript. I recommended to make a major revision prior to accepting it for publication.

1. The topic of the manuscript should be rephrased since the present topic is very similar to the published paper (Chemical Eng J, 2016,284:216-223) although their content is different.

Reply: Thank you very much for your suggestion. The topic has been rephrased in the revised manuscript.

2. L20-21, “The sludge DOM decreased to 272 mg dissolved organic carbon (DOC)/L with 65.2% reduction by Mucor sp. ZG-3 in 3 days” changed into “Sludge dissolved organic carbon (DOC) decreased to 272 mg /L with 65.2% reduction by Mucor sp. ZG-3 in 3 days”

Reply: Thank you very much for your valuable suggestion. The sentence has been revised as your suggestion in the revised manuscript.

3. L77-78, “390.85 mg dissolved organic carbon (DOC)/L” changed into “390.85 mg DOC/L”

Reply: Thank you very much for your suggestion. This related content has been revised in the revised manuscript.

4. L32, what means is d50 ? You should provide the full name of abbreviation when it appeared in the first time.

Reply: Thank you very much for your suggestion. D50 is median particle size, representing sludge average particle size, and the full name of d50 has been added in the revised manuscript.

5. L39, “findings” should be ”studies”

Reply: Thank you very much for your suggestion. The word “findings” has been replaced with the word “studies”.

6. L102-104, “…purposes: (1) investigated……; (2) evaluate……” should be “…purposes to (1) investigate……; and (2) evaluate…….

Reply: Thank you very much for your suggestion! This sentence has been revised in the revised manuscript.

7. L116, “specific resistance to filtration (SRF)” should be “SRF“

Reply: Thank you very much for your suggestion. The phrase “specific resistance to filtration (SRF)” has been replaced with the abbreviation “SRF”.

8. L118 and Table 1, delete “The solid sludge content”. I suggest to delete Table 1. The data in Table 1 can insert into the text.

Reply: Thank you very much for your suggestion. The phrase “The solid sludge content” and Table 1 have been deleted, and the data in Table 1 has been inserted into the text.

9. L183, please describes dialysis process in details including how many volumes of distilled water dialysis bag is placed into. Exchange times of distilled water? Does it need to shake or stir during dialysis?

Reply: Thank you very much for your valuable suggestion. 5 mL of DOM solution was added into dialysis bag and dialyzed against 1 L distilled water in beaker at 4°C. During dialysis, external solution was replaced with distilled water at intervals of 3 h for 12 times over a period of 2 days and the dialysis bag was fully shaken in exchanging process in order to remove the low molecular weights fraction. The dialysis process in details has been described in the revised manuscript.

10. Subtitle “Evaluation of the effect of sludge DOM with different MW on Fe2+ oxidation by A. ferrooxidans LX5” cannot stand for the content of the text. Please rephrase it. It seems to be that “The effect of different molecular weight DOM on Fe2+ oxidation by A.ferrooxidans LX5”.

Reply: Thank you very much for your helpful comment. The subtitle has been revised to “The effect of different molecular weight DOM on Fe2+ oxidation by A.ferrooxidans LX5”.

11. L194-212. What is DOM concentration obtained by autoclaving, centrifuging and filtering sludge? How do you get 572 mgDOC/L of sludge L-DOM? Doesn’t it need to freeze-drying dialyzed external solution and then re-dissolve it to obtain a given concentration of L-DOM? Likewise, what about other fractions of DOM with different MW? Please check it carefully and describe the procedures.

Reply: Thank you very much for your comment. In order to eliminate the effect of indigenous microorganisms on DOM, the sludge was firstly autoclaved, and the DOM concentration of autoclaved sludge is 782 mg DOC/L. After dialysis, the concentrations of DOM with MW<3000 Da and 3000 Da<MW<4000 Da were 549.8 mg DOC/L and 22.2 mg DOC/L, respectively, representing 572 mg DOC/L of sludge L-DOM (MW<4000 Da). The method of collection of sludge DOM with different MW by freeze-drying and re-dissolving is of great guiding significance for the development of our experiments. However, in order to adequate dialysis of sludge DOM, external solution need to be replaced with distilled water frequently and massively, resulting in the large amount of dialyzed external solution. It is a time-consuming process to freeze-drying these dialyzed external solution. In order to shorten collection time, 500 mL sludge DOM was dialyzed with dialysis bag of 4000 Da against 500 mL distilled water at 4°C. Dialyzed dialysis bag was replaced with new dialysis bag of 4000 Da containing fresh sludge DOM at intervals of 24 h for 3 times over a period of 3 days under the same external solution. After dialysis, the DOM in external solution was approximately sludge L-DOM. Because of osmotic pressure, distilled water will inflow into the dialysis bag resulting in the reduction of external solution, therefore the sludge L-DOM was obtained by diluting the external solution to the specified concentration. By similar dialysis operation, we obtained the sludge M-DOM and sludge H-DOM. Firstly, sludge DOM was dialyzed with dialysis bag of 4000 Da to remove the L-DOM fraction. Secondly, 500 mL dialyzed sludge DOM was dialyzed with dialysis bag of 14000 Da against 500 mL distilled water at 4°C. Dialyzed dialysis bag was replaced with new dialysis bag of 14000 Da containing fresh dialyzed sludge DOM without L-DOM at intervals of 24 h for 3 times over a period of 3 days under the same external solution. Finally, the sludge M-DOM was obtained by diluting the dialyzed external solution to the specified concentration. The collection of sludge H-DOM was only carried out by dialyzing sludge DOM with dialysis bag of 14000 Da against distilled water at 4°C. The external solution was replaced with distilled water at intervals of 3 h for 12 times over a period of 2 days, then the internal solution was sludge H-DOM. The procedure has been checked and described carefully in the revised manuscript.

12. L232, delete “Evaluation of”

Reply: Thank you very much for your suggestion. The phrase “Evaluation of” has been deleted in the revised manuscript.

13. There are too many grammar and syntax errors in the manuscript. Besides, Table and figures are also irregular.

Reply: Thank you very much for your suggestion. The manuscript has been carefully checked, and grammar and syntax errors have been corrected, and the formatting of table and figures have also been adjusted to regularization in the revised manuscript.

Response to the Reviewer #2:

Reviewer #2: The results were nicely presented. The following corrections needs to be addressed.

1. Page 12: equation 2 – mention the actual formula, whcih can make readers easy to understand. (not the values in formula)

Reply: Thank you very much for your suggestion. The equation 2 has been modified into the actual formula in the revised manuscript as suggested.

2. Page 19: Line 381 – the statement “It is L-DOM not H-DOM….” is little confusing. Better the statement can be rephrased for better understanding.

Reply: Thank you very much for your suggestion. The sentence has been rephrased for better understanding in the revised manuscript.

3. Fig 8 – Y axis – volume fraction cannot have %. It is a fraction.

Reply: Thank you very much for your suggestion. Fig 8 showed the particle size distribution of sludge. Y axis in figure represents volume fraction of sludge particle size, meaning the ratio of the content of a particular sludge particle size to that of all sludge particles. It is better to present the volume fraction with “%”.

4. Fig 6 – Inoculation at the end of day 1 – the vertical line is shown not at 1 day in X axis – Error needs to be corrected in both graph 6A and 6B.

Reply: Thank you very much for your suggestion. All vertical lines of “Inoculation at the end of day 1” have been moved to 1 day in X axis in relevant figures.

Response to the Reviewer #3:

Reviewer #3: 

1. Provide keywords at the end of abstract.

Reply: Thank you very much for your suggestion. According to the format of PLOS ONE, keywords are not required in the manuscript but have been entered in the keywords section of the submission system.

2. Research gap is essential between literature review and objectives. It is missing in the manuscript.

Reply: Thank you very much for your suggestion. Although sludge DOM only achieved 42.5% degradation to 390.85 mg DOC/L after 3 days of fungal treatment, which was still much higher than the inhibitory concentration (150 mg DOC/L ) to A. ferrooxidans LX5, the inhibition of degraded DOM to A. ferrooxidans LX5 was significantly alleviated. Sludge DOM encompasses fractions of varying molecular weights (MWs), and different MW DOM fractions may exert differential effects on Fe2+ oxidation by A. ferrooxidans LX5. However, there are few reports evaluating the effect of different MW DOM on Fe2+ oxidation by A. ferrooxidans LX5. Meanwhile, for engineering applications and cost savings, it is necessary to explore the optimal Fe2+ supplementation for sequential bioleaching process. Research gap has been added and emphasized in the revised manuscript.

3. Novelty statement is not clear.

Reply: Thank you very much for your suggestion. Although the concentration of degraded DOM by Mucor sp. ZG-3 was still high, the inhibition of degraded DOM to A. ferrooxidans LX5 was significantly alleviated. The alleviation mechanism of sludge DOM inhibition to A. ferrooxidans LX5 by Mucor sp. ZG-3 might be owing to the degradation of specific DOM which was probably mainly the inhibitory substances to A. ferrooxidans LX5. Therefore, the novelty of study is that the inhibition of different MW DOM to A. ferrooxidans LX5 and the exploration of optimal Fe2+ supplementation for sequential bioleaching process, which is a further extension to our previous studies. The novelty statement has been clearly emphasized in the revised manuscript.

4. Background, methods, results, discussion and conclusion are written very well.

Reply: Thank you very much for your comment. Language and context have been further refined and revised in the revised manuscript.

5. References section is not updated. Only 7 out of 44 is after 2020.

Reply: Thank you very much for your suggestion. References have been updated in the revised manuscript.

6. Authors may provide supplementary information within in the manuscript.

Reply: Thank you very much for your suggestion. All relevant data are provided within the manuscript and its Supporting Information files.

---

## [Editor Report · Decision Letter 1]

2 Apr 2024

Enhancing sludge dewaterability in sequential bioleaching: Degradation of dissolved organic matter (DOM) by filamentous fungus Mucor sp. ZG-3 and the influence of energy source

PONE-D-24-03890R1

Dear Dr. Wang,

We’re pleased to inform you that your manuscript has been judged scientifically suitable for publication and will be formally accepted for publication once it meets all outstanding technical requirements.

Kind regards,

Rakesh Namdeti

Academic Editor

PLOS ONE
---

## [Editor Report · Acceptance letter]

8 May 2024

PONE-D-24-03890R1 

PLOS ONE

Dear Dr. Wang, 

I'm pleased to inform you that your manuscript has been deemed suitable for publication in PLOS ONE. Congratulations! Your manuscript is now being handed over to our production team.

Kind regards, 

on behalf of

Dr. Rakesh Namdeti 

Academic Editor

PLOS ONE